# Immunomodulatory Properties of Polysaccharides from *Lentinula edodes*

**DOI:** 10.3390/ijms23168980

**Published:** 2022-08-11

**Authors:** Aleksander Roszczyk, Jadwiga Turło, Radosław Zagożdżon, Beata Kaleta

**Affiliations:** 1Department of Clinical Immunology, Medical University of Warsaw, Nowogrodzka 59, 02-006 Warsaw, Poland; 2Department of Drug Technology and Pharmaceutical Biotechnology, Medical University of Warsaw, Banacha 1, 02-097 Warsaw, Poland

**Keywords:** immunomodulation, *Lentinula edodes*, polysaccharide

## Abstract

*Lentinula edodes* (Berk.) Pegler, also known as shiitake mushroom, is a popular edible macrofungus and a source of numerous bioactive substances with multiple beneficial health effects. *L. edodes*-derived polysaccharides are the most valuable compounds, with anticancer, antioxidant, antimicrobial, and immunomodulatory properties. It has been demonstrated that their biological activity depends on the extraction method, which affects monosaccharide composition, molecular weight, branching degrees, and helical conformation. In this review, we discuss the immunomodulatory properties of various polysaccharides from *L. edodes* in animal models and in humans.

## 1. Introduction

*Lentinula edodes* (Berk.) Pegler, also known as shiitake mushroom, is the most popular edible macrofungus, belonging to the Basidiomycota [1]. *L. edodes* grows naturally in the humid and warm regions of North and South America, Asia, and Australia and is also widely cultivated all over the world [2]. Numerous studies have demonstrated that *L. edodes* fruiting bodies and mycelium are a source of various nutrients and bioactive substances with multiple beneficial health effects [2,3]. *L. edodes* contains about 68–78% of carbohydrates (mono-, di-, tri-, and polysaccharides), exogenous amino acids (including arginine, histidine, leucine, isoleucine, lysine, methionine, phenylalanine, threonine, valine, and tryptophan), lipids (5–8% of dry matter), vitamins (including B1, B2, B12, C, D, E), minerals (including Ca, K, Mg, Mn, P, Zn, and Na), fiber, and numerous substances with antibacterial, antiviral, and antifungal properties [3,4]. The main compounds of *L. edodes* are presented in Figure 1. *L. edodes*-derived polysaccharides are widely studied and the most valuable compounds with proven anticancer, antioxidant, antiaging, antimicrobial, and immunomodulatory properties [5,6]. Studies on the mechanism of the immunomodulatory action of fungal polysaccharides suggested that it is a complex interaction between immunological, metabolic, and epigenetic changes [7]. As detailed below, β-glucans are the main polysaccharides found in the fungal cell wall and are responsible for most of the biological effects. Glucan molecules are considered to be pathogen-associated molecular patterns (PAMPs) and various pattern-recognition receptors (PRRs) can recognize them (please see below for details) [7,8].

It is worth emphasizing that besides polysaccharides, other *L. edodes* compounds are biologically active [5,6]. Findings from recent studies indicated that exogenous amino acids regulate cytokines and nitric oxide (NO) synthesis, antibodies production, and lymphocytes proliferation [3]. In addition, evidence of B and D vitamins role in the immune response has accumulated in past years. It has been demonstrated that B vitamins regulate the function of T helper (Th) cells, natural killer (NK) cells, and macrophages, including cells proliferation, cytokines synthesis, and NO production. Vitamin D also regulates the function of all immune cells, including T and B lymphocytes, monocytes/macrophages, and dendritic cells. Minerals present in *L. edodes*, particularly Mg and Zn, are essential for the innate and adaptive immune response. They regulate the function of macrophages, neutrophils, T and B lymphocytes, as well as NK cells [4].

It has been demonstrated that cultivation and extraction method are essential steps in obtaining polysaccharides because they affect their monosaccharide composition, different molecular weight, branching degrees, helical conformation, as well as biological activity [7,8].

In this review, we discuss immunomodulatory properties of polysaccharides from *L. edodes* in animal models and in humans.

## 2. Polysaccharides from *L. edodes*—Immunomodulatory Properties

### 2.1. Surface Receptors and Signaling Pathways

Over the past three decades, numerous studies have shown that *L. edodes*-derived polysaccharides exhibit immunomodulatory properties. It has been confirmed that they can modulate both innate and adaptive immune responses, and thus should be considered as biological response modifiers (BRMs) [9,10]. *L. edodes* biologically active polysaccharides mostly belong to the group of β-glucans. Glucans are PAMPs, therefore numerous cell surface receptors on monocytes/macrophages, neutrophils, natural killer (NK) cells, dendritic cells (DCs), as well as T and B cells recognize them [11,12]. So far, several polysaccharide-binding receptors have been identified, including complement receptor 3 (CR3), dectin-1, and lactosylceramide (LaCer). CR3 is expressed on neutrophils, monocytes, macrophages, and NK cells and recognizes mainly β-1,3-glucans [11]. The interaction of polysaccharides with CR3 leads to enhanced spleen tyrosine kinase (Syk) phosphorylation, augmented phosphatidylinositol3 kinase (PI3K) activation, and stimulation of nuclear factor kappa B (NF-κB) leading to the cytokine production (including interleukin (IL)-2, IL-10, tumor necrosis factor (TNF)-α) [13]. Dectin-1 is expressed on monocytes, macrophages, neutrophils, T cells, and dendritic cells. It recognizes β-1,3- and β-1,6-glucans. Binding of β-glucans to dectin-1 activates numerous signaling pathways, including Syk, PI3K/protein kinase B (Akt), mitogen-activated protein kinase (MAPK), and nuclear factor of activated T cells (NFAT),to enhance innate immune responses (phagocytosis, reactive oxygen species poduction, IL-12, IL-6, IL-10, and TNF-α synthesis) [12,13]. Lactosylceramide (LaCer) is another polysaccharide-binding receptor which recognizes β-1,3-glucans. LaCer is highly expressed on neutrophils, macrophages, and dendritic cells. The interaction of polysaccharides with LacCer enhances neutrophil oxidative burst response, induces macrophage inflammatory protein (MIP)-2, and activates NF-κB (via protein kinase C (PKC)-dependent pathway) [13,14]. It has been suggested that fungal β-glucans can also be recognized by scavenger receptors (expressed on macrophages and dendritic cells), however, the exact mechanism of their activation is not fully understood. It is believed that ligand binding is associated with activation of kinases PI3K/Akt and MAPK [15,16]. β-glucan receptors on immune cells and immune activation induced by β-glucans are presented in Figure 2.

Due to the fact that fungal polysaccharides from varoius sources have different composition, molecular weight, and conformation, they can bind to multiple receptor types with different affinity and activate various signalling pathways. Therefore, it is very likely that future research will contribute to the discovery of new receptors for muschroom-derived polysaccharides.

### 2.2. Immunomodulatory Properties of Polysaccharides from L. edodes—Studies Performed in Animals

Numerous studies have been conducted to determine the immunomodulatory effects of polysaccharides isolated from mycelium or fruiting bodies of *L. edodes* in animal models.

Chen et al. [8] investigated immunomodulatory properties of three polysaccharide fractions (named F1, F2, and F3) isolated from *L. edodes* water extracts in BALB/c mice. All three fractions were composed of glucose (Glc), galactose (Gal), and mannose (Man) in different proportions. F1 was mainly composed of 38.0% Glc, 12.2% Gal, and 12.3% Man. F2 mainly consisted of 28% Glc, 4.6% Gal, and 4.4% Man; F3 contained 13.6% Glc, 2.7% Gal, and 2.5% Man. Moreover, all three fractions had different molecular weights (136 kDa, 14–61 kDa, and 14–35 kDa, respectively). The study demonstrated that F1, F2, and F3 polysaccharides increased thymus index and footpad thickness in mice. In addition, all three fractions increased proliferation of Concanavalin A- and LPS-stimulated splenocytes. The group also revealed that F2 and F3 upregulated serum immunoglobulin(Ig)G and IgM levels and enhanced NK cells cytotoxicity. These results suggest that F1, F2, and F3 polysaccharides isolated from *L. edodes* had immunoenhancing activity. The fraction with the highest molecular weight (F1) improved cellular immunity, while F2 and F3 improved both cellular and humoral immunity.

In a study by Pan et al. [17], the effect of supplementation of *L. edodes*-derived 1,3/1,6-β glucan on C57BL/6J mice gut microbiota was evaluated. It was reported that both short-term (7 days) and long-term (15 weeks) supplementation of β-glucan (at the dose of  1.5 mg per mouse/day) inhibited macrophage accumulation in the colon of mice fed by a high-fat diet and downregulated the expression of proinflammatory cytokines, including IL-6, TNF-α and IL-1β.

The immunomodulatory properties of bioprocessed polysaccharide (BPP) isolated from *L. edodes* liquid mycelial culture supplemented with black rice bran were studied by Kim and colleagues [18]. BPP is a β-glucan composed of glucose, galactose, rhamnose, fucose, mannose, and xylose. It was found that BPP upregulated spleen lymphocyte proliferation, as well as increased serum IFN-α levels in mice infected with *Salmonella* Typhimurium. In addition, BPP administration augumented Th1 cytokines production (including IL-1β, IL-2, IL-6, and IL-12) in mice splenocytes but did not affect Th2 cytokines production (IL-4, IL-5, and IL-10). The same polysaccharide, isolated from mycelial culture supplemented with tumeric reduced the phagocytic activity of the chicken-derived macrophage cell line HD-11 when infected with *Salmonella* Gallinarum and upregulated the expression of IL-1β, IL-10, TNF-α and inducible nitric oxide synthase in response to various *Salmonella* infections [19].

In another study, the biological activity of five novel polysaccharides (described as SLNT1, SLNT2, JLNT1, JLNT2, and JLNT3) isolated from the dried fruit body of *L. edodes* was evaluated in H22-bearing mice [20]. All polysaccharides consisted of glucose and had similar structures of (1→3)-linked-β-D-glucan and triple-helix conformation. Biological activity analyzes demonstrated that supplementation of two out of five tested polysaccharides with the highest molecular weights (SLNT1 and JLNT1) suppressed H22 tumor growth. Moreover, these two polysaccharides stimulated IL-2 and TNF-α production and induced the tumor cell apoptosis.

In a study by Song et al. [21], the biological effect of acidic-hydrolysis polysaccharides from spent mushroom compost of *L. edodes* (ASMCP) was evaluated in mice with LPS-induced kidney injury (KI). ASMCP is a homogenous β-glucan composed of rhamnose, (32.36%), arabinose (22.58%), galactose (8.92%), and glucose (36.14%), with molar ratios of 4:3:1:4. Immunological studies demonstrated that ASMCP reduced the serum levels of TNF-α, IL-6 and IL-1β, urea nitrogen, creatinine, and uric acid. Moreover, it upregulated activity of superoxide dismutase, catalase, and malondialdehyde, suggesting their antioxidant, antiinflammatory, and renoprotective activity.

In another study, the impact of lentinan, a (1→3)-β-glucan with (1→6) branches, encapsulated into CaCO_3_ microparticles on lymphocyte activation, as well as humoral and cellular immunity in BALB/c mice, was analyzed [22]. Animals were immunized with ovalbumin (OVA) and injected with lentinan (0.2 mL at a concentration of 10 μg/mL). It was demonstrated that lentinan-CaCO_3_-OVA enhanced lymphocyte proliferation, increased activation of B cells, and upregulated the ratio of CD4^+^ to CD8^+^ T cells in spleen lymphocytes. In addition, increased secretion of immunoglobulins IgG and IL-2, IL-4, IFN-γ, and TNF-α was observed.

In another similar research, lentinan was covalently attached to multiwalled carbon nanotubes (MWCNTs) and its immunoregulatory capacities were analyzed in BALB/c mice [23]. Hydrophobic MWCNTs in this study were used as an adjuvant in a vaccine delivery system for intensification of the immune response. The study demonstrated that lentinan attached to MWCNTs (L-MWCNTs) had higher solubility, lower cytotoxicity, and higher biostability than soluble lentinan. L-MWCNTs enhanced production of IL-4, IL-6, TNF-α, and IFN-γ. Moreover L-MWCNTs upregulated the percentage of CD4^+^ and CD8^+^ T cells in the spleen and enhanced the production of anti-OVA IgG antibodies. The observed enhancement of the immunomodulatory effects of lentinan was associated with the fact that nanotubes rapidly entered immune cells, contained larger amounts of antigens, and potentiated cellular and humoral immunity.

Wang and colleagues [24] investigated the effect of lentinan on the immune cells of BALB/c mice with sepsis. Animals received 40, 100 or 250 mg/kg of lentinan daily (by intraperitoneal injection). It was found that the drug (at all doses tested) significantly decreased IL-10 production and FoxP3 expression by CD4^+^CD25^+^ Tregs. Moreover, it attenuated LPS-stimulated Erk–FoxO1 activation. The obtained results led to the hypothesis that lentinan could be used as an immunoregulatory drug preventing sepsis in patients with burns, however, further studies in this field are needed.

The immunomodulatory effects of orally administered lentinan were also studied in rainbow trout (*Oncorhynchus mykiss*) with LPS-induced inflammation [25]. It has been demonstrated that lentinan decreased the expression of TNF- and IFN-related genes involved in acute inflammatory reactions.

In another study, the impact of lentinan supplemenation on immune response in Brown Norway rats was studied by McCormack et al. [26]. Lentinan (12 mg/kg) was orally administered for 20 weeks and the number of immune cells, as well as serum levels of GM-CSF, IFN-γ, IL-1α, IL-1β, IL-2, IL-4, IL-6, IL-10, and TNF-α, was determined. It was found that lentinan supplementation increased the number of circulating monocytes and CD8^+^ T cells and decreased the ratio of CD4/CD8 cells. Moreover, this drug reduced the production of IL-4, IL-6, IL-10, and GM-CSF.

Zheng et al. [27] isolated and purified a polysaccharide, named L-II from the fruiting body of *L. edodes*, and evaluated its effects on the cellular immune response in Sarcoma-180-bearing mice. The polysaccharide L-II consisted of d-glucopyranose and had molecular weight of 2.03 × 10^5^ Da. Animals were treated with L-II in a doses 1, 5, and 10 mg/kg body weight. It was demonstrated that L-II in all tested concentrations increased the thymus and spleen weight, delayed-type hypersensitivity response, as well as the phagocytosis of macrophages. Moreover, this polysaccharide elevated serum TNF-α and IFN-γ but not IL-2. In addition, L-II increased nitrogen oxide production and catalase activity in macrophages.

Jeff et al. [28], likewise, isolated and purified two mannogalactoglucan-type polysaccharides (WPLE-N-2 and WPLE-A0.5-2), from the fruiting bodies of *L. edodes*, and evaluated their effects on the cellular immune response of Sarcoma-180-bearing mice. Both polysaccharides were α-glucans composed of glucose, galactose, and mannose. Animals were treated with 100 mg/kg body weight of the polysaccharides for 10 days. The group observed a significant tumor regression in the polysaccharides’ groups. Moreover, polysaccharides upregulated NO production in peritoneal macrophages and macrophage phagocytosis and enhanced concanavalin and LPS-induced splenocytes proliferation.

Park and colleagues [29] investigated whether a standardized mycelial extract of *L. edodes*, containing α- and β-glucans, amino acids, and minerals (named AHCC^®^) promotes the therapeutic effect of immunotherapy in cancers in mice. The group used a combination of oral AHCC^®^ and dual immune checkpoint blockade (DICB), including PD-1/CTLA-4 blockade. It was demonstrated that the treatment reduced tumor growth and increased granzyme B and Ki-67 expression by tumor-infiltrating CD8^+^ T cells in MC38 colon-cancer-bearing mice compared with a combination of water and DICB.

All of the results, which are summarized in Table 1, suggest that polysaccharides isolated from *L. edodes* have immunomodulatory properties. However, different studies gave opposite results. Reasons for such divergences may be different animal models used in studies, different chemical structure and molecular weight of analyzed polysaccharides, or even lack of precise analysis of which of the components of the extracts/powdered *L. edodes* is responsible for the observed immunomodulatory activity. Therefore, future investigation in this field is needed.

### 2.3. Immunomodulatory Properties of Polysaccharides from L. edodes—Studies Performed in Humans

The first studies on the biological activity of *L. edodes* polysaccharides in humans were carried out in the 1980s.

The immune activating effects of lentinan were firstly described in healthy humans by Aoki [30] and Tani et al. [31]. It was found that this drug in vitro enhanced the proliferation of peripheral blood mononuclear cells (PBMCs) and increased the production of lymphokine-activated killer cel and NK cells activity. Similar results were obtained in gastric cancer patients when lentinan was administered intravenously (2 mg of lentinan for 7 days) with classic chemiotherapy [31,32]. Moreover, it was found that this drug significantly increased IL-1α, IL-1β, and TNF-α production by PBMCs of these patients, and increased the ratio of activated T cells and cytotoxic T cells in the spleen [30,32]. In another study, conducted in patients with gastric cancer the effect of lentinan on lymphocyte subsets of peripheral blood, lymph nodes, and tumor tissues was evaluated [33]. A total of 2 mg of lentinan was intravenously administered twice to twelve patients. No changes in the lymphocyte subsets of peripheral blood were found, however, the number of CD4^+^ T cells in lymph nodes was elevated. In addition, the number of tumor-inflitrating CD4^+^, Leu11, and LeuM3 cells was significantly increased.

The effect of orally administered lentinan on the immune response in healthy elderly humans was investigated in the placebo-controlled clinical study in 2011 [34]. Forty-two participants were randomly assigned to two groups given either 2.5 mg/day Lentinex^®^ or placebo. It was found that the number of total CD3^+^ T cells and CD4^+^ T cells was lower in the placebo group than in the Lentinex^®^ group, however, no changes in the number of CD8^+^ T cells were detected. In addition, it was demonstrated that the number of B cells in the placebo group was nonsignificantly decreased and nonsignificantly increased in the Lentinex^®^ group. Interestingly, the difference between groups after six weeks after the end of the supplementation was significant. The number of CD56^+^ NK cells increased significantly in both groups, but there was no difference between the groups. Changes in humoral response were also described. The complement C3 decreased significantly in both groups, but no difference between the groups was observed. No statistically significant changes in IgG, IgA, and IgM levels were found between the studied groups.

In another randomized, placebo-controlled clinical trial, the hypocholesterolemic, microbiota-modulatory, and immunomodulatory effect of *L. edodes* extract was analyzed in hypercholesterolemic patients [35]. Fifty-two participants were assigned to two groups given either a β-D-glucan-enriched (BGE) mixture (10.4 g/day) obtained from *L. edodes* fruiting bodies ensuring a 3.5 g/day of fungal β-D-glucans or a placebo incorporated in three different food products. No significant differences in lipid- or cholesterol-related parameters were found compared with the placebo group. In addition, the BGE showed no immunomodulatory effect (including changes in IL-1β, IL-6, and TNF-α production). However, BGE modulated the colonic microbiota.

The immunomodulating properties of *L. edodes* mushroom have also been investigated in healthy young adults in another randomized study [36]. Fifty-two participants were randomized to consume either one (5 g) or two (10 g) servings of dried, prepared *L. edodes* daily for 4 weeks. It was demonstrated that mushrooms consuming increased ex vivo proliferation of γδ-T cells and NK-T cells, as well as the expression of activation receptors on these cells. In addition, an increased level of secretory IgA in saliva and decreased level of C-reactive protein in serum was observed. The concentration of IL-1α, IL-1β, IL-4, IL-6, IL-10, IL-17, IFN-γ, TNF-α, macrophage inflammatory protein-1α/chemokine C-C ligand 3 (MIP-1α/CCL3), and MIP-1β in PBMCs culture supernatants was also analyzed. It was found that mushroom intake upregulated IL-4, IL-10, TNF-α, and IL-1α levels and downregulated MIP-1α/CCL3. No changes were observed for IL-6, IL-1β, MIP-1β, IL-17, and IFN-γ. It was concluded that regular *L. edodes* consumption resulted in improved immunity, however, it was not analyzed which component of the mushroom exerted immunomodulatory effects.

Wang et al. [37] evaluated the influence of lentinan on cytokines profile and T cell subsets in 73 patients with nonsmall cell lung cancer (NSCLC) treated with vinorelbin and cisplatin. As expected, all patients with NSCLC had a higher subset ratio of CD3^+^CD8^+^ T cells. It was found that the combined therapy of lentinan–vinorelbin–cisplatin resulted in a significant increase in CD3^+^ CD56^+^ NKT cells, CD3^+^CD8^+^, and CD3^+^CD4^+^ subsets. Moreover, therapy with lentinan decreased the number of CD4^+^CD25^+^ Tregs, increased the production of IL-10 and TGF-β1, and upregulated IFN-γ, TNF-α, and IL-12. It was concluded that lentinan-based chemo-immunotherapy could be a promising strategy in NSCLC treatment.

Our group also investigated the immunomodulatory properties of polysaccharides from *L. edodes* in healthy human immune cells. We isolated from the mycelial cultures a selenium(Se)-enriched analog of lentinan (named Se-L) and, as a reference, the non-Se-enriched fraction (named L) and evaluated its effects on the proliferation of human PBMCs stimulated with various T and B-cells mitogens and influence on superoxide anions (O_2_^−^) production by granulocytes [38]. Our study demonstrated that none of the polysaccharides has a significant effect on the generation of reactive oxygen species by granulocytes. However, we found that both fractions significantly decreased the proliferation of PMBCs stimulated with anti-CD3 antibody and phytohemagglutinin (PHA) in a dose-dependent manner. Structural studies of the obtained fraction revealed that it is a protein-containing mixture of high molar mass polysaccharides (MW 3.9 × 10^6^ g/mol and 2.6 × 10^5^ g/mol) α- and β-glucans composed of glucose or mannose. At this early stage of the research, it was not determined which of the components of the fraction was responsible for the immunosuppressive activity. Therefore, in our next study, we separated homogenous (Se)-enriched fraction (named Se-Le-30), defined four-polysaccharide components of this fraction, and determined their structures and immunomodulatory properties in human immune cells [39,40]. We found that Se-Le-30 downregulated the proliferation of human PBMCs stimulated with anti-CD3 antibodies or allostimulated and decreased the production of TNF-α by CD3^+^ T cells. This result suggests that Se-Le-30 fraction is a selective T cells immunosuppressant, which presumably acts through the modulation of signaling via the TCR–CD3 receptor complex. Later analyses demonstrated that all four polysaccharide components of Se-Le-30 also inhibited the proliferation of anti-CD3-stimulated PBMCs, but the inhibitory effect was weaker than that of the Se-Le-30 fraction. In continuation of our study, we extracted from a postculture medium of *L. edodes* exopolysaccharides fraction (Se-containing and nonselenated) composed mainly of a highly branched 1-6-α-mannoprotein of molecular weight 4.5 × 10^6^ Da and also evaluated its effect on human PBMCs and granulocytes. The analysis demonstrated that similarly to polysaccharides isolated from *L. edodes* mycelium, exopolysaccharides have no influence on the production of reactive oxygen species by granulocytes but reduce PMBCs proliferation induced by the anti-CD3 antibody. However, the observed effect was much weaker than that for the fractions isolated from mycelium [41].

Most of the results (summarized in Table 2) strongly suggest that polysaccharides from *L. edodes* have immunomodulatory properties in humans, however, various polysaccharide fractions may significantly differ in biological activity. These differences result both from the polysaccharides source (fruiting bodies or mycelium) and extraction method, which affects monosaccharide composition, molecular weight, branching degrees, and helical conformation [7,8]. Lentinan, a (1→3)-β-glucan with (1→6) branches, extracted from fruiting bodies of *L. edodes*, functions as an effective immunostimulatory agent. In contrast, polysaccharides with different structure isolated from *L. edodes* mycelium have selective immunosuppressive activity.

## 3. Polysaccharides from *L. edodes* as Vaccine Adjuvants

A separate area of research is the analysis on the use of *L. edodes* polysaccharides as vaccine adjuvants in order to increase the immunogenicity of antigens [42]. It has been proven that polysaccharides of natural origin are safe, nontoxic, have good biocompatibility, and are able to stimulate the immune response alone [43].

Lentinan as a biologically active polysaccharide was investigated as a potential adjunct for Bacillus Calmette–Guérin (BCG) vaccination [44]. Drandarska and colleagues studied the immunomodulating effects of the combined application of BCG with lentinan (intranasal application at a dose of 1 mg/kg, three times at 2-day intervals) in guinea pigs. Forty-five days after treatment, H_2_O_2_ and nitrite production by alveolar macrophages, as well as their killing ability against *Mycobacterium tuberculosis* and *Staphylococcus aureus*, were measured. The obtained results suggest that BCG plus lentinan application induced a higher level of alveolar macrophage activation and enhanced the local immunohistological response to BCG in lungs.

In another study, it was suggested that lentinan enhanced the effects of the vaccine against *Trichinella spiralis* in C57BL/6J mice [45]. Recombinant *T. spiralis* Serpin (rTs-Serpin) was used as a vaccine with lentinan as an adjuvant. It was demonstrated that this combination reduced the rate of helminth burden, increased the levels of anti-rTs-serpin IgG1 and IgG2a antibodies, and upregulated production of IFN-γ and IL-4 by CD4^+^ T cells. Moreover, the group investigated the role of NOD-, LRR- and pyrin domain-containing protein 3 (NLRP3) in vaccine + lentinan-induced immune protection and found that lentinan increased the expression of NLRP3 mRNA, suggesting that this transcription factor may play a role in lentinan-induced immunoprotection.

Since only few studies have been conducted to analyze the potential of lentinan to influence vaccine effectiveness, further research is urgently needed. When testing new polysaccharides as vaccine adjuvants, it is important to take into consideration their separation methods, structure, and purity.

## 4. Concluding Remarks and Future Perspectives

In the last three decades, the immunomodulatory effects of various polysaccharides from *L. edodes* have been described. As summarized in the present review, numerous in vitro and in vivo studies conducted on animal models and in humans suggested that these glucans can be used as immunostimulants or immunosuppressants in many conditions.

Research confirms that *L. edodes* polysaccharides can modulate an immune response by various biochemical pathways, however, some analyses gave inconclusive or conflicting results. Reasons for such divergences may be the presence of many variables that affect studies effects, including different *L. edodes* strains used, different cultivation conditions, various isolation and purification methods, as well as different dosage and route of administration.

Therefore, future studies must be focused not only on understanding the immune-related pathways and mechanisms but also on an improvement in the methodological quality.

## Figures and Tables

**Figure 1 ijms-23-08980-f001:**
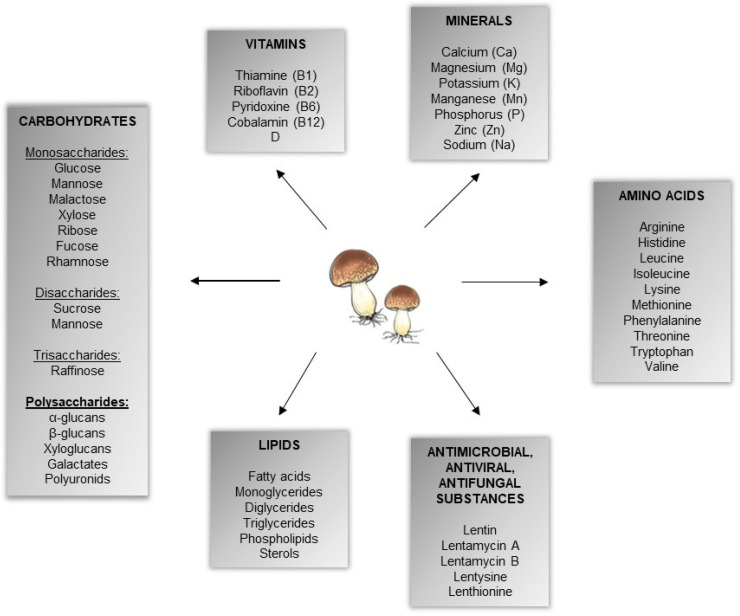
Main active compounds of *Lentinula edodes*.

**Figure 2 ijms-23-08980-f002:**
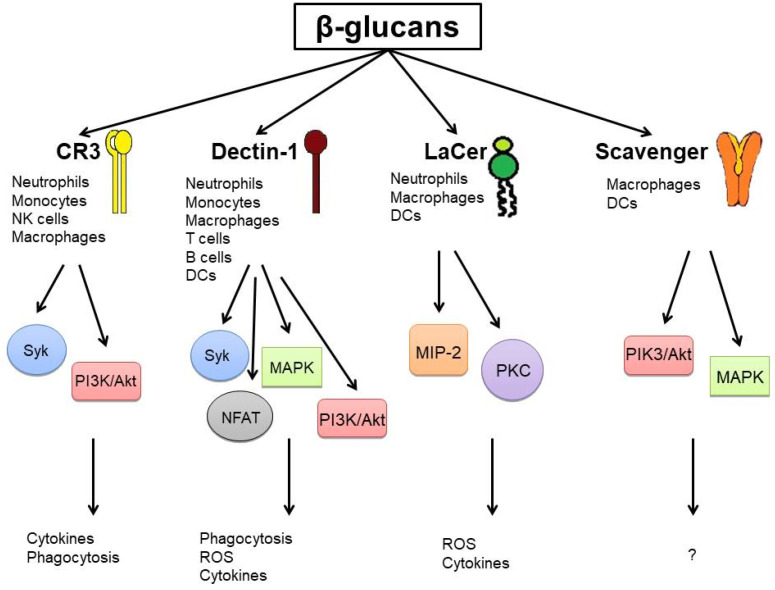
Main β-glucan receptors on immune cells and their immunomodulatory properties.

**Table 1 ijms-23-08980-t001:** Main studies of immunomodulatory effects of *L. edodes* polysaccharides in animal models.

Sample	Animal Model	Immunomodulatory Effects	Reference
F1, F2, and F3 heteropolysaccharide fractions composed of glucose, galactose, and mannose in different proportions, isolated from *L. edodes* water extracts	BALB/c mice	1. Increased thymus index and footpad thickness (F1, F2, and F3 fraction). 2. Increased proliferation of Con A- and LPS-stimulated splenocytes (F1, F2, F3 fraction). 3. Increased serum IgG and IgM levels (F1, F2, F3 fraction). 4. Enhanced cytotoxic activity of NK cells (F2 and F3 fraction).	Chen et al. [8]
1, 3/1, 6-β-glucan from *L. edodes*	C57BL/6J mice	1. Inhibited macrophage accumulation in the colon. 2. Inhibited expression of proinflammatory cytokines: IL-6, TNF-α, and IL-1β.	Pan et al. [17]
Bioprocessed polysaccharide (BPP)—β-glucan composed of glucose, galactose, rhamnose, fucose, mannose, and xylose, isolated from *L. edodes* liquid mycelial culture supplemented with black rice bran	BALB/c mice infected with *Salmonella* Typhimurium	1. Upregulated spleen lymphocyte proliferation. 2. Increased serum IFN-α levels. 3. Increased IL-1β, IL-2, IL-6, and IL-12 production in splenocytes.	Kim et al. [18]
Bioprocessed polysaccharide (BPP)—β-glucan composed of glucose, galactose, rhamnose, fucose, mannose, and xylose, isolated from *L. edodes* liquid mycelial culture supplemented with tumeric	Chickens infected with *Salmonella* Gallinarum	1. Reduced phagocytic activity of the chicken-derived macrophage cell line HD-11. 2. Increased expression of IL-1β, IL-10, TNF-α, iNOS. 3. Reduced expression of IL-4, IL-6, IFN-β, IFN-γ.	Han et al. [19]
SLNT1, SLNT2, JLNT1, JLNT2, and JLNT3 polysaccharides ((1→3)-β-glucans composed of glucose), isolated from *L. edodes* fruit body	H22-bearing mice	1. Suppressed H22 tumor growth and enhanced tumor cell apoptosis (SLNT1 and JLNT1). 2. Increased IL-2 and TNF-α production (SLNT1 and JLNT1).	Wang et al. [20]
Polysaccharide (homogenous β-glucan composed of rhamnose, arabinose, galactose, and glucose) isolated from *L. edodes* spent compost (ASMCP)	Kunming strain mice with LPS-induced kidney injury	1. Reduced the serum levels of TNF-α, IL-6 and IL-1β. 2. Reduced serum levels of urea nitrogen, creatinine, and uric acid. 3. Increased activity of superoxide dismutase, catalase, and malondialdehyde.	Song et al. [21]
Lentinan ((1→3)-β-glucan with (1→6) branches) encapsulated into CaCO_3_ microparticles	BALB/c mice	1. Enhanced lymphocyte proliferation. 2. Enhanced activation of B cells. 3. Increased ratio of CD4^+^ to CD8^+^ T cells in spleen lymphocytes. 4. Increased serum IgG levels. 5. Increased IL-2, IL-4, IFN-γ, and TNF-α production.	Liu et al. [22]
Lentinan covalently attached to multiwalled carbon nanotubes	BALB/c mice	1. Enhanced production of antiovalbumin IgG antibodies. 2. Increased production of IL-4, IL-6, TNF-α, and IFN-γ. 3. Upregulated percentage of CD4^+^ and CD8^+^ T cells in spleen	Xing et al. [23]
Lentinan	BALB/c mice	1. Decreased IL-10 production. 2. Decreased FoxP3 expression of CD4^+^CD25^+^ Tregs. 3. Decreased LPS-induced IL-10 production in Tregs. 4. Attenuated LPS-stimulated Erk–FoxO1 activation.	Wang et al. [24]
Lentinan	Rainbow trout (*Oncorhynchus mykiss*) with LPS-induced inflammation	1. Decreased expression of TNF- and IFN-related genes.	Djordjevic et al. [25]
Lentinan	Brown Norwegian rats	1. Increased number of circulating monocytes. 2. Increased number of CD8^+^ T and decreased ratio of CD4/CD8 cells. 3. Decreased production of IL-4, IL-6, IL-10, and GM-CSF.	Jeff et al. [26]
L-II polysaccharide consisted of D-glucopyranose, isolated from *L. edodes* fruiting body	Sarcoma 180-bearing mice	1. Increased the thymus and spleen weight. 2. Increased delayed-type hypersensitivity response. 3. Increased phagocytosis of macrophages. 4. Increased TNF-α and IFN-γ serum levels. 5. Increased NO production and catalase activity in macrophages.	Zheng et al. [27]
WPLE-N-2 and WPLE-A0.5-2 polysaccharides(α-glucans composed of glucose, galactose, and mannose), isolated from *L. edodes* fruiting body	Sarcoma 180-bearing mice	1. Increased NO production in peritoneal macrophages. 2. Increased phagocytosis of macrophages. 3. Enhanced Con-A and LPS-induced splenocytes proliferation.	Jeff et al. [28]
AHCC^®^, a standardized extract of cultured *L. edodes* mycelia containing α- and β-glucans, starches, sugars, amino acids, and minerals	C57BL/6 mice	1. Reduced tumor growth. 2. Increased granzyme B and Ki-67 expression by tumor-infiltrating CD8^+^ T cells.	Park et al. [29]

Abbreviations: Con-A, concanavalin A; GM-CSF, granulocyte-macrophage colony-stimulating factor; IFN, interferon; IL, interleukin; NK, natural killer; NO, nitrogen oxide; LPS, lipopolysaccharide; TNF, tumor necrosos factor.

**Table 2 ijms-23-08980-t002:** Main studies of immunomodulatory effects of *L. edodes* polysaccharides in humans.

Sample	Participants	Immunomodulatory Effects	Reference
Lentinan used in PBMCs culture	Healthy humans	1. Enhanced proliferation of PBMCs. 2. Increased production of lymphokine-activated killer cell. 3. Increased NK cells activity.	Aoki [30] Tani et al. [31]
Lentinan administered intravenously with chemiotherapy	Gastric cancer patients	1. Enhanced proliferation of PBMCs. 2. Increased production of lymphokine-activated killer cell. 3. Increased NK cells activity. 4. Increased IL-1α, IL-1β, and TNF-α production by PBMCs. 5. Increased ratio of activated T cells and cytotoxic T cells in the spleen. 6. Elevated number of CD4 cells in lymph nodes 7. Increased number of tumor-inflitrating CD4, Leu11, and LeuM3 cells.	Tani et al. [31] Arinaga et al. [32] Takeshita et al. [33]
Lentinan administered orally	Healthy eldery humans	1. Higher number of total CD3^+^ and CD4^+^ T cells. 2. Increased number of B cells (nonsignificantly).	Gaullier et al. [34]
β-D-glucan-enriched mixture obtained from *L. edodes* fruiting bodies incorporated in food products	Patients with hypercholesterolemia	1. No changes in IL-1β, IL-6, and TNF-α production.	Morales et al. [35]
Whole dried *L. edodes*	Healthy young humans	1. Increased ex vivo proliferation of γδ-T cells and NK-T cells. 2. Increased expression of activation receptors on γδ-T cells and NK-T cells. 3. Increased level of secretory IgA in saliva. 4. Decreased level of CRP in serum. 5. Upregulated production of IL-4, IL-10, TNF-α, and IL-1α by PBMCs. 6. Downregulated production of MIP-1α/CCL3 by PBMCs.	Dai et al. [36]
Lentinan in combined therapy with vinorelbin and cisplatin	Patients with non-small cell lung cancer (NSCLC)	1. Increased number of CD3^+^CD56^+^ NKT cells. 2. Increased number of CD3^+^CD8^+^ T cells. 3. Increased number of CD3^+^CD4^+^ T cells. 4. Derceased number of CD4^+^CD25^+^ Tregs. 5. Increased production of IL-10, TGF-β1, IFN-γ, TNF-α, and IL-12.	Wang et al. [37]
Polysaccharides (Se-L and L) isolated from *L. edodes* mycelium used in PBMCs and granulocytes cultures	Healthy humans	1. Decreased proliferation of anti-CD3 Ab and PHA-stimulated PBMCs. 2. No effects on production of O_2−_ by granulocytes.	Kaleta et al. [38]
Homogenous polysaccharide fraction (Se-Le-30) isolated from *L. edodes* mycelium used in PBMCs and granulocytes cultures	Healthy humans	1. Decreased proliferation of anti-CD3 Ab-stimulated PBMCs. 2. Decreased proliferation of allostimulated PBMCs. 3. Decreased production of TNF-α by CD3^+^ T cells. 4. No effects on production of O_2-_ by granulocytes.	Kaleta et al. [39,40]
Exopolisaccharide isolated from a postculture medium of *L. edodes*	BALB/c mice	1. Decreased proliferation of anti-CD3 Ab-stimulated PBMCs. 2. No effects on production of O_2-_ by granulocytes.	Górska-Jakubowska et al. [41]

Abbreviations: CRP, c-reactive protein; IFN, interferon; Ig, immunoglobulin; IL, interleukin; MIP-1α/CCL3; macrophage inflammatory protein-1α/chemokine C-C ligand 3; NK, natural killer; PBMCs, peripheral blood mononuclear cells; PHA, phytohaemagglutinin; Se, selenium; TGF, transforming growth factor; TNF, tumor necrosos factor.

## Data Availability

The data presented in this study are available on request from the corresponding author without any restrictions.

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
