# Peer review of "Immunomodulatory Properties of Polysaccharides from Lentinula edodes"

_ijms, 2022, doi:10.3390/ijms23168980_

Round 1

Reviewer 1 Report

Fig. 1 is the key review guide point for a non specialized reader and therefore needs to be a little bit more descriptive and accurate. Concerning box "carbohydrates" - ribose name is misspelled and manganates definitely do not belong there. An info concerning aminoacids, provided in the introduction is not adequate. Since carbohydrates constitute by far the major portion of shiitake extracts, well balanced comments on possible immunomodulatory action (and potency) of the minor components (like AA) seem appropriate.

Author Response

RESPONSES TO REVIEWER’S 1 COMMENTS:

We carefully considered Reviewer’s 1 comments. We want to extend our appreciation for taking the time and effort necessary to provide such insightful guidance.

Herein, we explain how we revised the manuscript entitled: „ Immunomodulatory properties of polysaccharides from Lentinula edodes” based on those comments and recommendations. We hope that our revisions improve the paper such that Reviewer and Editor now deem it worthy of publication in IJMS. Below please find the detailed responses to Reviewer’s comments.

Reviewer’s 1 Comments and Suggestions for Authors:

Fig. 1 is the key review guide point for a non specialized reader and therefore needs to be a little bit more descriptive and accurate. Concerning box "carbohydrates" - ribose name is misspelled and manganates definitely do not belong there. An info concerning aminoacids, provided in the introduction is not adequate. Since carbohydrates constitute by far the major portion of shiitake extracts, well balanced comments on possible immunomodulatory action (and potency) of the minor components (like AA) seem appropriate.

Responses:

Dear Reviewer, we fully agree with all your comments and recommendations. As suggested, in the revised manuscript, we have we changed the Fig. 1. We have corrected the mistake in the word “ribose”, and removed manganates from the box “carbohydrates”. Thank you for paying attention to this subject. Moreover in the revised paper we have added more information about L. edodes amino acids and we have listed them in the “amino acids” box in the Fig. 1.

In addition, as suggested, in the Introduction we have mentioned about the possible immunoregulatory activity of other than polysaccharides compounds, including aa, vitamins, minerals. Moreover, according to Reviewer’s 2 recommendations, we have elaborated the Introduction in the context of immunomodulatory properties of polysaccharides. We hope that this will improve the quality of the manuscript. Thank you for these valuable suggestions.

Again, we appreciate all of your insightful comments. We worked hard to be responsive to them.

Author Response

RESPONSES TO REVIEWER’S 2 COMMENTS:

We carefully considered Reviewer’s 2 comments. We want to extend our appreciation for taking the time and effort necessary to provide such insightful guidance.

Herein, we explain how we revised the manuscript entitled: „ Immunomodulatory properties of polysaccharides from Lentinula edodes” based on those comments and recommendations. We hope that our revisions improve the paper such that Reviewer and Editor now deem it worthy of publication in IJMS. Below please find the detailed responses to Reviewer’s comments.

Reviewer’s 2 Comments and Suggestions for Authors:

In this review manuscript, various polysaccharides from L. edodes immunomodulatory properties were extensively discussed in animal and human models. The authors need to address all the points carefully as mentioned below

  1. Authors need to elaborate the introduction part more in the context of the immunomodulatory behavior of polysaccharides.
  2. Over the past three decades, numerous studies have shown that L. edodes-derived 42 polysaccharides exhibit immunomodulatory properties. It has been confirmed that they 43 can modulate both, innate and adaptive immune response, thus should be considered as 44 biological response modifiers (BRMs) [9,10].

How and which factors of polysaccharides modulate the immunomodulatory properties?

  1. In another similar research, lentinan was covalently attached to multiwalled carbon 139 nanotubes and its immunoregulatory capacities were analyzed in BALB/c mice. Authors need to discuss the detailed mechanism of the carbon nanotube / Lentinan for immunomodulatory properties modulation.
  2. How this review manuscript is different from the previously published articles?

Responses:

Dear Reviewer, we fully agree with all your comments and recommendations.

  1. As suggested, in the Introduction we have described the mechanisms of the immunomodulatory action of fungal polysaccharides, however, a detailed description of receptors and signaling pathways is provided in section 2.1 of the manuscript. Thank you for paying attention to this subject. Moreover, according to Reviewer’s 1 recommendations, in the Introduction we have mentioned about the possible immunoregulatory activity of other than polysaccharides compounds, including amino acids, vitamins, and minerals.
  2. In the Introduction (lines 33 – 39) and in section 2.1 we explain which factors of polysaccharides and how modulate the immune response. Thank you for the constructive remark.
  3. As requested, in the revised paper, we have discussed the mechanism of the carbon nanotube/lentinan immunomodulatory properties. Thank you for this valuable suggestion.
  4. Dear Reviewer, our manuscript summarizes the recent knowledge on the immunomodulatory properties of polysaccharides from edodes. Despite the fact that there are several review papers describing active compounds of this mushroom, many new studies in the past years have been conducted and numerous original papers have been published. In our manuscript we cited in detail studies conducted in animal models and in humans (described the structure of polysaccharides and immunomodulatory effects). Moreover, we discuss the use of L. edodes polysaccharides as vaccine adjuvants in order to increase the immunogenicity of antigens. Among all the papers cited, four are from 2021 and three from 2022. Therefore we believe that our review paper is valuable.

Again, we appreciate all of your insightful comments. We have taken your recommendations on board to improve and clarify the manuscript.

This manuscript is a resubmission of an earlier submission. The following is a list of the peer review reports and author responses from that submission.